# Soil and Freshwater Bioassays to Assess Ecotoxicological Impact on Soils Affected by Mining Activities in the Iberian Pyrite Belt

**DOI:** 10.3390/toxics10070353

**Published:** 2022-06-28

**Authors:** Óscar Andreu-Sánchez, Mari Luz García-Lorenzo, José María Esbrí, Ramón Sánchez-Donoso, Mario Iglesias-Martínez, Xabier Arroyo, Elena Crespo-Feo, Nuria Ruiz-Costa, Luis Roca-Pérez, Pedro Castiñeiras

**Affiliations:** 1Departamento de Biología Celular y Biología Funcional, Facultad de Ciencias Biológicas, Laboratorio de Ecotoxicología y Calidad Ambiental-LEyCA, Universitat de València, 46100 Valencia, Spain; oscar.andreu@uv.es (Ó.A.-S.); ruizcos@alumni.uv.es (N.R.-C.); 2Departamento de Mineralogía y Petrología, Facultad de Ciencias Geológicas, Universidad Complutense de Madrid, 28040 Madrid, Spain; jesbri@ucm.es (J.M.E.); ramons02@ucm.es (R.S.-D.); marioi01@ucm.es (M.I.-M.); ecrespo@ucm.es (E.C.-F.); castigar@ucm.es (P.C.); 3Centro de Asistencia a la Investigación de Técnicas Geológicas, Universidad Complutense de Madrid, 28040 Madrid, Spain; xarroyo@ucm.es; 4Departamento de Biología Vegetal, Facultad de Farmacia, Área de Edafología y Química Agrícola, Universitat de València, 46100 Valencia, Spain; luis.roca@uv.es

**Keywords:** soil contamination, potentially toxic elements, *Daphnia magna*, *Raphidocelis subcapitata*, earthworms, risk assessment, cytotoxicity

## Abstract

This study aims to use geochemical, mineralogical, ecotoxicological and biological indicators for a comprehensive assessment of the ecological risks related to the mobility, ecotoxicity and bioavailability of potentially harmful elements in the Lousal mining district. Particularly, toxicity was evaluated using four assays: algae, cytotoxicity assays with HaCaT cell line (dermal), earthworms and *Daphnia magna*. The geochemical and mineralogical characterization of the studied area shows that the mine wastes underwent intense weathering processes, producing important contamination of the adjacent soils, which also led to the release and mobilization of PHEs into nearby water courses. Total PTE results indicate that the soils affected by mining activities were highly contaminated with As and Cu, while Zn and Pb content ranged from low to very high, depending on the analyzed samples. Cadmium levels were found to be very low in most of the soil samples. The test using *Daphnia magna* was the most sensitive bioassay, while the *Eisenia foetida* test was the least sensitive. Except for the LOS07 soil sample, the rest of the soils were classified as “High acute toxicity” and “Very high acute toxicity” for aquatic systems. The results in HACaT cells showed results similar to the ecotoxicological bioassays. The application of biotests, together with geochemical and mineralogical characterization, is a very useful tool to establish the degree of contamination and the environmental risk of potentially harmful elements.

## 1. Introduction

Potentially harmful elements (PHEs) pollution arising from mining and smelting activities is a major environmental issue [1]. The major risk is the entering of these PHEs into the trophic chain and the ultimate effect on human beings. 

Surely, the process that produces the more intense impact on the environmental compartments is acidity production by acid mine drainage (AMD) [2]. This process occurs when sulfides are exposed to atmospheric, hydrological or biological weathering (oxygen, water and bacteria), becoming oxidized products and producing in the surficial water and groundwater, low pH and high concentrations of PHEs ions and sulfates [3]. This environmental concern is especially relevant in areas suffering from historical metallic mining activities, where the application of less efficient mineral extraction and processing techniques, and the lack of environmental awareness and controls have left huge environmental liabilities [4]. This is the case in the Iberian Pyrite Belt (IPB), where approximately 150 years of modern mining has resulted in the generation of enormous volumes of metal-rich waste dumps [5,6,7].

In order to assess the toxicity of PHEs in the different environmental compartments, the total element content does not provide enough information, because it does not reflect the ecotoxicological danger in the environment [8]. To estimate the environmental risk of contaminants, chemical methods need to be complemented with biological procedures [9]. 

Bioassays can provide relevant data on not only (bio)availability, but also on the effects on the trophic chain [10]. Therefore, ecotoxicological testing may be a useful approach for assessing toxicity as a complement to chemical analysis [11]. Any strategy for assessing properties potentially hazardous to the environment used in a classification system should include test organisms of different trophic levels, representing both the terrestrial and aquatic compartments [8].

The aim of this study was to check the effectiveness of a battery of bioassays in the screening of environmental risks in areas affected by mining activities and contaminated by PHEs, facilitating their zonation as a function of toxicity. Particularly, toxicity was evaluated using four assays: algae, cytotoxicity assays with HaCaT cell line (dermal), earthworms and *Daphnia magna.* Moreover, the possible relationship between observed toxicity and results of chemical analysis was studied. 

The suitability of those assays for the assessment of environmental risk in mine sites strongly polluted by PTEs is discussed and evaluated to determine if they could be applied in areas with similar contamination.

## 2. Materials and Methods

### 2.1. Study Area

The Lousal pyrite sulfide mine was active from 1900 to 1988, producing sulfur for fertilizers. The ore deposit is in the Alvalade Basin, in the western sector of the Iberian Pyrite Belt. The most common soils are leptosols [12], characterized by an acid character, with pH values around 5.2, and low organic matter content. In some areas, poorly developed soils over the Alvalade basin detritic deposits are also present, characterized by a sandy texture, very limited organic matter content and a slightly acidic nature [7].

The volcanic-sedimentary complex (VSC) at Lousal mine comprises from lower to upper terms: quartzite sequence/rhyolitic lavas/black and gray mudstone with ore sulfide deposits/rhyolitic sill/pillow basaltic units. The ore deposit is located near this VSC which has provided the hydrothermal circulation that led to ore formation. The ore deposit has a low metal content [13].

After extraction, the mineral was mechanically concentrated prior to sulfuric acid production. In the higher production period (the 1950s and 1960s), the Lousal mine had an annual production capacity of 200,000 to 250,000 tonnes. 

As usually occurs with mines that have had a long period of exploitation, a large volume of mining waste had been produced in Lousal, much of which is very reactive. Currently, two large volumes of these mine waste dumps remain in the abandoned mining exploitation, one derived from the production of the ore grinding plants and the other consisting of a large deposit of fine-grained pyrite located in an elevated sector of the mining area, in close proximity to the transport route. An ore deposit with pyrite as the main mineral phase can be considered a threat for their acidity production risk, and in Lousal, a huge environmental problem arose when mining production ceased in 1988. After this date, restoration and remediation actions were needed to solve the problem.

This remediation program developed a passive treatment composed of two actions: the encapsulation of the pyrite wastes and the remediation of the acid waters. The encapsulation was carried out with a geotextile and a layer of soil, while the chemical remediation consisted of the construction of two types of wetlands, aerobic and anaerobic, for the sequential treatment of acidic waters, with the aim of increasing the pH and favoring the precipitation of insoluble compounds of PHEs before discharge into the Corona River [14]. Nowadays, a European-funded mine reclamation project (LIFE RIBERMINE, LIFE ENV/ES/000181) is being applied in the soils affected by acid mine drainage (AMD) processes. 

### 2.2. Sampling Design

Samples used in this study correspond to a selection of soils characterized by Sánchez-Donoso et al. [7]. Selection criteria were based on contrasting PHE concentrations. As can be seen in Figure 1, the area has a dump with pyrite waste upstream and an AMD passive treatment (aerobic wetland) for the acid waters. Samples were collected from the mine soils in the restoration area before the works began. As Figure 1 displays, most of the samples (LOS06-LOS10) were taken in the west transect, in the flow direction downstream of the pyrite wastes (dump in Figure 1), while the last sample (LOS13) corresponds to the lower one in the East transect, close to the aerobic wetland treatment.

### 2.3. Geochemical Analysis

Major and trace elemental data were obtained by means of energy-dispersive X-ray fluorescence spectrometry (EDXRF), using a Malvern PANalytical, mod. Epsilon1 spectrometer. Sample preparation included grinding in an automatic agate mortar for 2 min until reaching a grain size less than 100 µm. Analysis was performed in four different energy ranges for 23 min to ensure that standard deviations were minimum. Quality control of the analyses included the analysis of duplicate samples and certified reference materials (SRM 2711) to check precision (Zn-87.3%, As-82.9%, Pb-93.1%, SO_3_-93.5%, Fe_2_O_3_-88.4%, Cu-85.0%) and accuracy (recovery percentages (in %): S (178.7–182.3), Fe (97.7–98.0), Cu (115.2–122.2), Zn (98.8–99.9), As (79.1–83.9), Pb (111.0–112.7)).

### 2.4. Mineralogical Analysis

Main mineral phases were determined by means of X-ray diffraction (XRD) using a Bruker D8 Advance diffractometer equipped with a Cu anticathode at the Unidad de Técnicas Geológicas from Universidad Complutense de Madrid. An aliquot of each soil was ground in an automatic agate mortar and manually, until all the grain size was less than 53 µm. Working conditions of the diffractogram obtention were: 2θ angles (2–68°), 0.02 stepping intervals and 1 s per step. The semiquantitative analysis was carried out according to the Chung method [15,16,17] using XPowder software (Ver 12, J Daniel Martín, Granada, Spain).

### 2.5. Soil Sample Leaching Procedure

A leaching process was conducted in each soil sample in order to obtain the aqueous extract used in the aquatic bioassays. The followed method was in accordance with Spanish legislation on contaminated soils, the Royal Decree (RD) 9/2005, which establishes the list of potentially polluting soil activities and the criteria and standards for the declaration of contaminated soils. The leaching method proposed by the RD is the DIN 38414-S4 [18], in which 100 g of the dry mass of the soil sample (with a particle size < 10 mm) is mixed in 1000 mL deionized water (10 L/S) and submitted to an upside-down agitation in a Reax20^®^ rotary agitator (Heidolph™, Schwabach, Germany, for 24 h at room temperature. The solid and liquid phases are separated by sedimentation at 4 °C overnight. The supernatant was then filtered in 0.45 μm pore size glass fiber membranes (Pall^®^) and stored at 4 °C in the dark until assayed.

### 2.6. Bioassays

#### 2.6.1. Acute Immobilization Test with Daphnia Magna

The biological material in the form of dormant eggs (ephippia) was supplied with a commercial kit, Daphtoxkit F™ (Microbiotests Inc., Ghent, Belgium). The assays were conducted in accordance with the OECD Guideline 202 [19] and summarized in the Standard Operating Procedure supplied by the manufacturer [20]. Only daphnids aged less than 24 h were used. Twenty neonates per concentration, plus one control were exposed in Daphtoxkit™ multiwell plates. Each assay was conducted in triplicate. The plates were incubated in the dark at 21 ± 1 °C. In order to assure the correct assay development, K_2_Cr_2_O_7_ was used as a positive control. After 24 h and 48 h of exposure, neonates’ immobilization was checked, and they were considered immobile if after 48 h of incubation with the toxicant they remained settled at the bottom of the test container and did not resume swimming within the observation period of 15 s. The tested dilutions were 50%, 25%, 12.5%, 6.25% and 3.12% (*v*/*v*). The EC_50_ was determined as the sample dilution required to immobilize 50% of the daphnids exposed after 48 h exposure.

#### 2.6.2. Freshwater Algae, Growth Rate Inhibition Test with *Raphidocelis Subcapitata*

The Algaltoxkit F™ (Microbiotest Inc., Ghent, Belgium) system was used in the study. The kit is in compliance with ISO 8692:2012 and OECD TG 201 [21]. The *R. subcapitata* cells (formerly *Pseudokirchneriella subcapitata*) were immobilized in alginate beads in accordance with the OECD guideline (Annex 3, TG 201). Five dilutions: 50, 25. 12.5, 6.25 and 3.12 (% *v*/*v*) and a negative control (100% of growth medium) with three replicates of each concentration were tested. The assays were conducted in 100 mm long cell cuvettes used in spectrophotometry. The initial biomass concentration of algae cells was adjusted to 10^4^ cells/mL. The long cells were incubated at 25 °C +/− 1 °C for 72 h under 4000 lux of continuous illumination in a climatic chamber, model ARTI-150L (Microbiotest Inc., Ghent, Belgium), equipped with four white light tone lamps. Growth inhibition rates relative to negative controls were determined by measurements of optical density (OD_670_) in a spectrophotometer model Auris 2021 (CECIL Instruments™, Cambridge, UK) equipped with a holder for cells of 10 cm path-length. The inhibition (% *I*) in the tested concentrations versus the control growth is based on the determination of the average growth rates (*µ*) after transformation of the OD_670_ values into cell numbers according to the manufacturer protocol [22]. The toxicity was expressed as 72 h ErC_50_.

#### 2.6.3. Earthworm Mortality Test

Assays were conducted following the OECD TG 207 for testing chemicals, [23] with minor adaptations [24]. *Eisenia foetida* earthworms were supplied by a local hatchery (Lombriventa, Spain) and acclimated to experimental conditions in polyethylene boxes and fed with a 1:1 mixture of horse manure and peat. Only adult specimens with differentiated clitellum (weight between 300 and 600 mg) were selected for the tests. Ten organisms were placed in a single-use food-grade plastic container (175 × 150 × 70 mm) containing 1 kg (dry weight, dw) of the prepared test sample soil. The test was conducted with artificial soil prepared according to OECD TG 207 (i.e., 70% silica sand, particle size between 80–120 µm, 20% kaolin clay and 10% Sphagnum peat) and the final content of organic matter was 2%. The moisture was adjusted with deionized water to 40% of water holding capacity and the pH corrected to 7.5 ± 0.5 by adding CaCO_3_ to the mixture in all test items including controls. Different soil sample percentages were added to the artificial soil to achieve nominal concentrations of 50%, 25%, 12.5, 6.25 and 3.12 (% *w*/*w*, dw). Each treatment was performed under continuous light (1000 lux) at a temperature of 20 ± 2 °C. Earthworm mortality and weight loss were evaluated on days 7 and 14. 

#### 2.6.4. Cytotoxicity Study Viability Assay

The HaCaT cell line, a non-tumorigenic immortalized human keratinocyte cell line, was obtained from the Central Service for Experimental Research (University of Valencia, Spain). Cells were cultured at 37 °C in a 5% CO_2_ humidified atmosphere in DMEM medium, supplemented with 10% fetal bovine serum (FBS), 2 mM L-glutamine, 100 U/mL penicillin, 100 μg/mL streptomycin and 250 μg/mL fungizone. Cell cultures were observed daily under an inverted phase-contrast microscope for morphology, growth and confluence control. For each experiment, cells were grown for 24 h in 96-multiwell flat-bottom plates (Corning^®^) and then the medium was replaced with fresh medium containing the dilutions: 50, 25, 12.5 and 6.25 (% *v*/*v*). The maximum dose of leachate was in accordance with the maximum dose of Milli-Q water that did not cause significant cytotoxicity in the controls (100% of cultured medium). Dilutions were made with DMEM free of FBS. Cell viability was determined after 24 h of exposure by the colorimetric 3-(4,5- dimethyl-2-thiazolyl)-2,5-diphenyl tetrazolium bromide (MTT) assay [25]. Briefly, 20 μL of MTT (5 mg/mL) in phosphate-buffered saline (PBS, pH 7.4) was added to each well and incubated for 3.5 h at 37 °C, 5% CO_2_. Medium containing leachate dilutions was then removed and 100 μL of 0.1 mM HCl in isopropanol was added to each well for solubilization of formazan crystals. The optical density of reduced MTT was measured at 590 nm and 620 nm (as reference filter) in a MultiSkanGo^®^ microtiter plate reader (Thermo Fisher Scientific Inc., Waltham, MA, USA). The inhibitory effect level was expressed as the leachate concentration that inhibits 50% of the exposed cell (IC_50_).

### 2.7. Ecotoxicological Statistical Analysis 

Whenever possible the toxicity was expressed as the effective concentration of the sample that gave half-maximal response (IC_50_/ErC_50_/EC_50_), along with 95% confidence limit values. The EC_50_ was determined by probit regression [26] implemented by the software EPA-Probit (v1.5). The soil risk characterization was based on the EC_50_ values and converted to Toxic Units (TU) according to the procedure proposed by Persoone et al. [27]. One-way analysis of variance (ANOVA) was carried out followed by Tukey’s post hoc analysis after testing normality and homoscedasticity of data distributions. Statistical analysis was done using SPSS™ (v21 for MS Windows™, IBM, Armonk, NY, USA).

## 3. Results and Discussion

### 3.1. Geochemical Analysis

Major and trace element content in selected samples is summarized in Table 1. The As content ranged from 254 to 530 mg kg^−1^, with maximum values in samples LOS13, and Pb ranged from 60 to 591mg kg^−1^ (maximum value in sample LOS06, at the beginning of the west transect). Other PHEs that showed noteworthy concentration values were Cu (up to 1320 mg kg^−1^ in sample LOS6) and Zn (up to 546 mg kg^−1^ in sample LOS06). The high Fe_2_O_3_content, with concentrations reaching 13% in three samples (LOS06, LOS09 and LOS10) and Co (more than 700 mg kg^−1^ in LOS06, LOS09 and LOS10), should also be mentioned [7].

Mineralogical data acquired shows that the main phases are quartz (26–60%), phyllosilicates (30–47%, mainly mica-illite and chlorite) and feldspars (8–14%), with goethite (Fe^3+^O(OH)), jarosite/plumbojarosite (KFe^3+^_3_(SO_4_)_2_(OH)_6_ and Pb0.5Fe^3+^_3_(SO_4_)_2_(OH)_6_). Alunite (KAl_3_(SO_4_)_2_(OH)_6_) and gypsum (CaSO_4_·2H_2_O) were identified as accessory phases only in sample LOS13 (Table 2). Secondary minerals (jarosite/plumbojarosite, alunite and goethite) produced by weathering processes affecting iron sulfide ores were identified in all samples. Primary ore minerals such as pyrite were not identified by X-ray diffraction, probably due to the oxidizing processes occurring in the uppermost meters of the soils in the studied area. One of the findings that should be highlighted is the high secondary gypsum content found in the LOS13 sample and the higher jarosite/plumbojarosite and alunite content in some of the samples (LOS06, LOS09, LOS10 and especially LOS13). Besides this, the concentrations of soluble secondary sulfates (jarosite, plumbojarosite and alunite) in the samples were considerably low (below 9%), suggesting the presence of active dissolution and mobilization processes of sulfates in the study area.

Since most of the bioassays were performed with the leachates of these soils, a complete characterization of them has been carried out. The PHEs with leachable concentrations below the detection limit are Pb and Cd, while there are significant amounts of leachable Cu and Zn in samples LOS06, LOS09, LOS10 and LOS13. Sample LOS07 only shows a very low amount of leachable Zn (8.1 mg L^−1^), therefore a low toxicity behavior of this sample in bioassays is expected (Table 3). The As extracted exhibited low mobility rates under natural conditions.

### 3.2. Ecotoxicological Assays

Terrestrial and aquatic ecotoxicological bioassays were successfully applied to soil and leachate samples. Toxicity (48 h EC_50_) and values converted to toxic units (T.U.) obtained in *D. magna*, *R. subcapitata* and *E. foetida* assays are noted in Table 4. 

All aqueous extract samples showed high toxicity to *Daphnia magna* except for the LOS07 soil sample. The order of toxicity of the samples was as follows (from higher to lower toxicity): LOS10 > LOS13 > LOS6 > LOS9 > LOS7. In the case of the test with the algae *R. subcapitata*, the order of toxicity was similar: LOS13 > LOS10 > LOS06 > LOS09 > LOS07. In both cases, the most toxic soil samples were LOS13 and LOS10.

The toxic effect on earthworms seems to be related to the higher bioaccumulation capacity of Zn, As, Cu and Pb [28]. Zn and Pb were found to be toxic to earthworms in studies carried out with mining soils containing these trace elements [29]. On the other hand, the possible additive effect that the presence of different metals such as Zn and Co may have on toxicity, which has already been reported for *D. magna* [30], should not be ruled out.

The test based on *E. foetida* only was sensitive at higher concentration tests (50%) for which LOS06, LOS07 and LOS10 had 100% mortality. However, LOS09 and LOS13 showed a mortality of 40% and 10% respectively. In all cases, the mortality occurred at 7 d of exposure. The earthworm’s weight was also checked, including controls at 0 d, 7 d and 14 d of exposure, according to the OECD TG 207 guideline. The most significant weight losses with respect to control for all exposure times (Tukey’s, *p* < 0.05) were found in LOS06. In LOS13, a major difference was detected in the concentration of 50% at 7 d and 14 d (Figure 2). For the rest of the samples, no differences were found between controls and soil different dilutions. In this case, the EC_50_ was calculated as the arithmetic mean between the two concentrations. Regarding the earthworm weight evolution during the study, there was a loss of fresh biomass in all concentrations, including controls.

Differences in sensitivity were appreciated, depending on test endpoints and organisms.

The most sensitive organism was *D. magna* followed by *R. subcapitata*. The terrestrial bioassays with earthworms were the less sensitive organism, this rank of sensibility has been confirmed in other ecotoxicological studies conducted in contaminated mining areas [24,28,31] and may be related to the earthworms’ avoidance behavior in the presence of stressors. The ecotoxicological evaluation of aqueous extracts from sampled soils confirmed sample LOS07 as the less toxic.

Chemical analysis showed that only Zn, Cu and As were found in the water extracts, and the rest of the trace metals present in whole soil samples were not detected. Both Zn and Cu were in concentrations above the EC_50_ values established for *R. subcapitata* and *D. magna* (Table 5).

The results were in accordance with metal concentrations obtained in industrial soils and from mining areas contaminated by trace metals [29]. Trace elements such as Zn can increase its toxicity (additive effect) by the presence of other trace metals (i.e., Co), this can explain the toxicity in the elutriates when only Zn and Cu were detected [30,40]. Since Copper (Cu) is a microelement essential for algae metabolism, it is involved in many physiological processes, acting as cofactors in many enzymes such as Cu/Zn superoxide dismutase, oxidative phosphorylation or in iron mobilization [41]. Nevertheless, in higher concentrations, such as in collected samples, it can exert a toxic effect [42].

Two methods for the classification of soil according to its toxicity have been applied: (i) The method developed by Persoone et al. [27], based on a hazard classification system for waste discharged into the aquatic environment with five toxicity classes: from “No acute toxicity” (Class I) to “Very high acute toxicity” (Class V); (ii) The Spanish Royal Decree 9/2005, which establishes in its Annex III the criteria and standards for the declaration of contaminated soils. The RD establishes as “contaminated soils” those with an EC_50_ < 1%. In accordance with RD Annex IV, if the protection of the ecosystem is considered a priority by the local authorities—i.e., declared a protected natural area—an environmental risk assessment should be carried out.

According to the RD 9/2005, all soil samples were classified as contaminated soils except for LOS07 for the *D. magna* test. Regarding the *R. subcapitata* results, sample LOS13 was the only one classified as contaminated. For earthworms, all of the soil samples were considered to be contaminated.

With respect to the hazard classification system for wastes discharged into the aquatic environment [27], the soil elutriates were classified as follows: LOS07 had a “slight acute toxicity” for *D. magna* and *R. subcapitata*, LOS09 had “high acute toxicity” and the rest of samples (LOS06, LOS10 and LOS13) were classified as “very high acute toxicity”. Considering the results from the ecotoxicological tests on whole soils and soil leachates, the highest toxicity was obtained in the LOS13 soil sample, followed by LOS10 = LOS06, LOS09 and the sample site LOS07 had the lowest toxicity.

The dermic route, in conjunction with the respiratory and ingestion, are the most significant routes of human exposure to potentially harmful elements (PHE). Cytotoxicity studies can be used to measure the potential hazard posed by exposure to toxic substances as an alternative to in vivo experiments. The HaCaT cell line is derived from human immortalized keratinocytes and is generally used in skin barrier toxicity assays. Its use is unusual in the environmental risk assessment of contaminated mining soils and the available information in this regard is scarce. Nevertheless, there are in vitro studies based on other cell lines (Caco-2, HL-7702) used to evaluate the bioaccessibility, bioavailability or absorption of PHE and trace elements from contaminated soils [43,44,45]. The cell viability inhibition (MTT assay) conducted with aqueous extracts exposed to HaCaT cells is shown in Figure 3.

The aqueous extract from LOS06 was the most cytotoxic. The cytotoxic effect was statistically significant with respect to the control (*p* < 0.05). LOS07 and LOS09 did not show cytotoxic effects; no differences with respect to the control were found. Finally, LOS10 and LOS13 only showed a cytotoxicity effect at the higher exposed concentration (10%) where the difference with respect to the control was significant. There is dose–response between the trace element concentration in leachates (Table 2) and cytotoxicity (Figure 3). The greatest cell viability inhibitions were found in elutriates with the highest presence of trace elements, especially Zn, which is regarded as a dangerous cation, its toxic effect in the cell systems exceeding that of iron, copper, manganese and cobalt in the same concentration range [46]. The in vitro cytotoxicity with HaCaT keratinocytes and human in vivo data was well-correlated by [47] and has also been proven as an alternative method for assessing skin irritancy and toxicity [48], in this regard, LOS06, LOS10 and LOS13 mining soils may pose a risk if the soil dust or soil particles make contact with persons or animals.

## 4. Conclusions

The Lousal mining area was exploited from 1900 to 1988, resulting in the production of a large volume of mine waste. The geochemical and mineralogical characterization of the studied area shows that the mine wastes underwent intense weathering processes, producing important contamination of the adjacent soils, which also led to the release and mobilization of PTEs into nearby water courses. Total PTE results indicate that the soils affected by mining activities were highly contaminated with As and Cu, while Zn and Pb content ranged from low to very high, depending on the analyzed samples. Cadmium levels were found to be very low in most of the soil samples.

The ecotoxicity bioassays performed in selected soils showed different sensitivity to the target contaminants. Regarding PHE concentrations, a greater toxic effect on earthworms could be expected, however, the contact test performed with *Eisenia foetida* suggested that this test is the least sensitive of the four bioassays applied.

Even if the total PHE content is high, the aqueous extracts presented a much lower PHE load, with only Zn being detected (five times lower in the extract than in the soil), Cu (between 10 and 20 times lower) and As (more than 1000 times lower in the aqueous extract than in the soil). However, the leachates showed high toxicity, the test with *Daphnia magna* being the most sensitive, followed by the test with the algae *Raphidocelis subcapitata*. The lack of sensitivity of the *E. foetida* test can be explained by the ability of earthworms to detect the presence of contaminants and avoid them. This should be taken into account to propose other types of endpoints based on reproduction or behavior (avoidance-behavior).

The use of alternative methods for the assessment of toxicity in higher organisms is widely recommended by the European Union. In this sense, this study has used cultures of immortalized keratinocyte cells in order to evaluate the potential effect on human health through the dermal entry route. The results show a similar relationship between the levels of metals detected in the elutriates of each soil and cytotoxicity, to that found in direct ecotoxicological bioassays.

Apart from the LOS07 soil sample, the rest of the soils were classified as “high acute toxicity” and “very high acute toxicity” for aquatic systems. This fact must be taken into account when proposing actions aimed at restoring the study area due to the possible effect on the receiving channels or the underlying aquifers.

The application of ecotoxicity bioassays, together with geochemical and mineralogical characterization, is a very useful tool for establishing the degree of contamination and the environmental risk of PHEs. In addition, it is necessary to have environmental regulations delimiting the degree of soil contamination through ecotoxicological criteria; in this sense, RD 9/2005 establishes at which degree of toxicity (EC_50_) a soil is considered to be contaminated (EC_50_ < 1%).

## Figures and Tables

**Figure 1 toxics-10-00353-f001:**
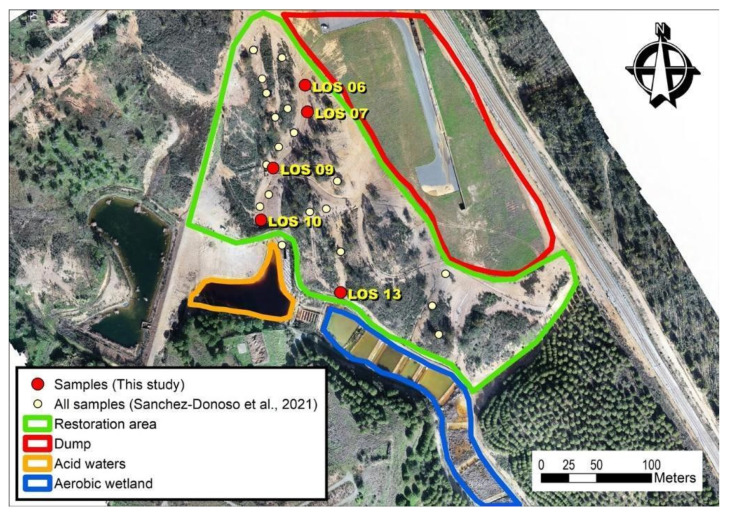
Location of samples in the restoration area.

**Figure 2 toxics-10-00353-f002:**
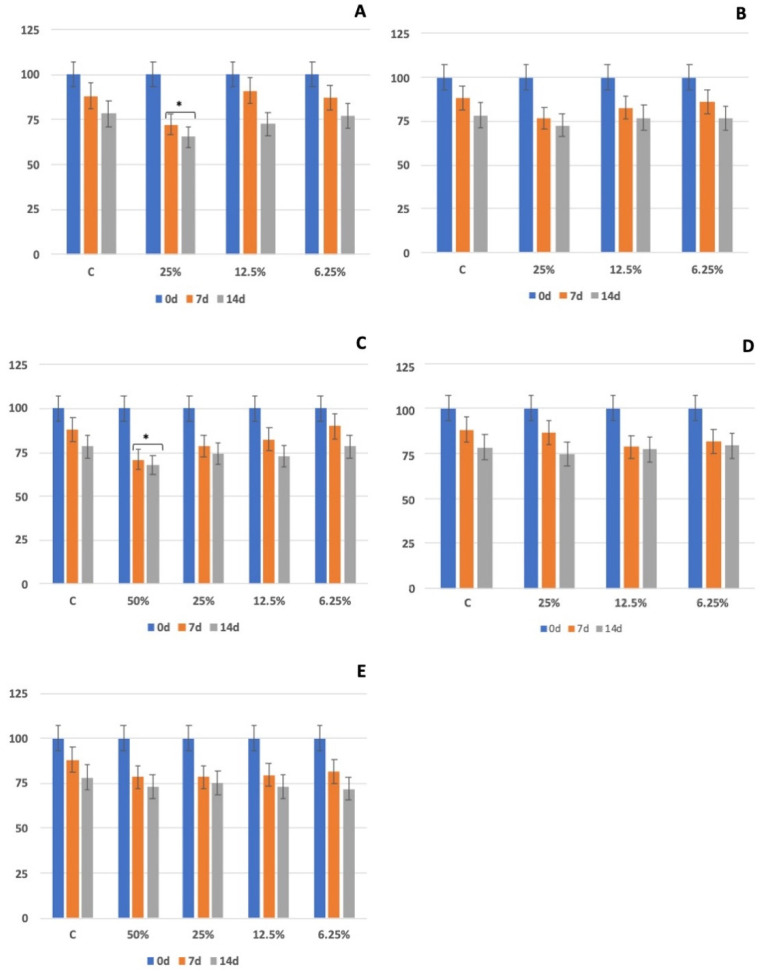
Earthworm weight variation at t = 0.7 and 14 days of exposure (mean ± SD, *n* = 10). *X*-axis: Soil concentration (%, *w*/*w*). *Y*-axis: Earthworm weight variation (%). Analysis of variance (ANOVA) was performed between the control and treatment groups followed by Tukey’s post hoc test. * *p* < 0.05, is considered to be statistically significant compared with control. Sample points: (**A**): LOS06, (**B**): LOS07, (**C**): LOS09, (**D**): LOS10, (**E**): LOS13.

**Figure 3 toxics-10-00353-f003:**
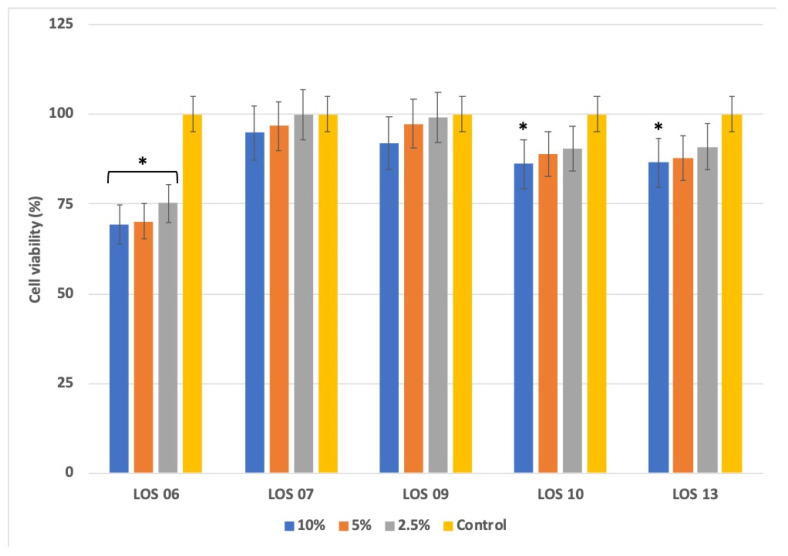
Human HaCaT cell toxicity in soil leachates and control. *X*-axis: Leachate concentration (% *v*/*v*) in each soil sample site: *Y*-axis: Cell viability expressed as percentage (%). Values indicate the mean ± SD in three independent experiments. The ANOVA has been made between “% leachate” vs. Control inside each sampling point. Asterisk indicates significant difference in HaCaT cell viability between samples and control (Tukey’s post hoc test * *p* < 0.05).

**Table 1 toxics-10-00353-t001:** Major and trace element concentrations in soil samples.

Sample	LOS 06	LOS 07	LOS 09	LOS 10	LOS 13
Al_2_O_3_ (%wt)	21.9	18.7	25.2	23.4	24.1
SiO_2_ (%wt)	53.8	63.4	54.1	54.2	51.2
P_2_O_5_ (%wt)	0.5	0.5	0.5	0.5	0.5
SO_3_ (%wt)	3.3	3.1	1.4	2.2	6.0
Cl (%wt)	0.1	0.2	0.1	0.1	0.1
K_2_O (%wt)	5.3	4.3	5.2	5.1	4.9
Ti (%wt)	0.7	0.6	0.6	0.7	0.6
Fe_2_O_3_ (%wt)	13.2	8.7	12.2	13.2	9.6
CaO (%wt)	0.5	0.3	0.2	0.2	2.5
V (mg kg^−1^)	184	123	199	207	187
Cr (mg kg^−1^)	137	81	145	134	120
Mn (mg kg^−1^)	389	552	507	353	528
Co (mg kg^−1^)	812	581	714	769	590
Ni (mg kg^−1^)	68	55	72	61	71
Cu (mg kg^−1^)	1320	365	318	512	432
Zn (mg kg^−1^)	546	163	308	367	539
Ga (mg kg^−1^)	27	30	34	36	30
As (mg kg^−1^)	530	421	312	430	254
Rb (mg kg^−1^)	285	210	230	257	231
Sr (mg kg^−1^)	145	148	121	121	156
Y (mg kg^−1^)	46	50	31	42	54
Zr (mg kg^−1^)	504	442	294	441	421
Nb (mg kg^−1^)	26	27	19	25	21
Sn (mg kg^−1^)	77	54	65	78	67
Te (mg kg^−1^)	30	35	37	36	30
Ba (mg kg^−1^)	396	253	331	409	336
Pb (mg kg^−1^)	591	61	105	140	167
Eu (mg kg^−1^)	120	66	55	105	68
Yb (mg kg^−1^)	64	47	42	59	45

**Table 2 toxics-10-00353-t002:** Mineralogical composition (wt%) of the studied surficial samples. Qtz: quartz; Msc: muscovite; Ill: illite; Chl: chlorite; Mnt: montmorillonite; Fsp: feldspars; Gt: goethite; Jar: jarosite; Alu: alunite; Gyp: gypsum.

	Qtz	Msc	Ill	Chl	Mnt	Fsp	Gt	Jar	Alu	Gyp
LOS 6	49	35		1	4	9	1	1		
LOS 7	60	30				8	2			
LOS 9	38	32		9	4	14	2	1		
LOS 10	26	33	14	7	4	11	4	1		
LOS 13	30	32		8		10	1	2	5	12

**Table 3 toxics-10-00353-t003:** Water leachable content of some PHEs in the studied samples.

PHE	Soil Sample Sites				
	LOS06	LOS07	LOS09	LOS10	LOS13
As (mg/L)	0.011	0.006	0.004	0.003	0.01
Zn (mg/L)	152.2	16.2	80.4	113.3	144.3
Cu (mg/L)	99.6	0.1	22.6	31.3	27.7

**Table 4 toxics-10-00353-t004:** 48 h EC_50_ for *D. magna* assays and 72 h ErC_50_ for *R. subcapitata* assays (95% confidence intervals) expressed as percentage of water extract in test medium (*v*/*v*). 14 d EC_50_ in *E. foetida* assays expressed as percentage of dw soil in medium (*w*/*w*).

Aqueous Extracts Toxicity Tests		Sample Sites				
		LOS 06	LOS 07	LOS 09	LOS 10	LOS 13
*D. magna* immobilization	48 h EC_50_	0.66 (0.61–0.70)	NT	1.0 (0.56–1.8)	0.34 (0.30–0.39)	0.5 (0.42–0.76)
	TU *	152	<1	99	290	179
	Hazard Class [27]	V	I	IV	V	V
	R.D. 9/2005 classification **	C	NC	C	C	C
*R. subcapitata* Inhibition growth	72 h ErC_50_	3.02 (2.1–5.5)	NT	11.6 (5.8–37.3)	1.3 (1–1.8)	0.1 (0.05–0.2)
	TU *	33.1	<1	8.6	78.1	1000
	Hazard Class [27]	IV	I	III	IV	V
	R.D. 9/2005 classification **	NC	NC	NC	NC	C
Whole soil toxicity test						
*E. foetida* mortality	14 d EC_50_	37.5 (NOEC: 25%)	33.9 (NOEC: 25%)	>50 (NOEC: 25%)	37.5 (NOEC: 25%)	> 50 (NOEC: 50%)
	TU *	2.7	2.9	<2	2.7	<2
	Hazard Class [27]	III	III	II	III	II
	R.D. 9/2005 classification **	NC	NC	NC	NC	NC

NT: No Toxic. NOEC: No Observed Effect Concentration. TU: Toxic Units (100/EC_50_). NT: No Toxic. No Observed Effect Concentration (NOEC) = 100%. (EC_50_ >100%). * Class: TU < 0.4 Class I (No acute toxicity); 0.4 < TU < 1 Class II (Slight acute toxicity); 1 < TU < 10 Class III (Acute toxicity; 10 < TU < 100 Class IV (High acute toxicity); TU >100 Class V (Very high acute toxicity). ** R.D. 9/2005 (Annex III) Classification: EC_50_ < 1% C (Contaminated); EC_50_ > 1% N.C. (Not Contaminated).

**Table 5 toxics-10-00353-t005:** EC_50_ values for Zinc, Copper and Arsenic in *D. magna* (mg/L), *R. subcapitata* (mg/L), *E. foetida* (mg/kg) and Cytotoxicity (µM or mg/L) to HaCaT cell line if not otherwise stated.

Trace Element	Bioassay			
	*D. magna* (48 h EC_50_)	*R. subcapitata* (72 h ErC_50_)	*E. foetida* (14 d EC_50_)	HaCaT (24 h IC_50_)
Zn	0.82 [32]	0.1 [32]	NOEC: 100 [33]	35.6 [34] (A549 cells) 33.5 [34] (HEK cells
Cu	0.21–0.44 [35]	0.03–0.82 [36]	8.4 [37]	NOEC: 580 µM [38] (36.9 mg/L)
As	25.2 [37]	1.5 [37]	413 [37]	4.8 µM [39] (0.36 mg/L)

## Data Availability

Mari Luz García-Lorenzo, as responsible of the manuscript entitled “Soil and freshwater bioassays to assess ecotoxicological impact in Soils Affected by Mining Activities in the Iberian Pyrite Belt”, authored by myself and Andreu-Sánchez, O, Esbrí, JM, Sánchez-Donoso, R, Iglesias-Martínez, M, Arroyo, X, Crespo-Feo, E, Ruiz-Costa, N, Roca-Pérez, L. and Castiñeiras, P. On behalf of the rest of the coauthors, with this document I warrantee and sign that the datasets generated and used during the current study are available from the corresponding author on reasonable request.

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
