# Peer review of "Soil and Freshwater Bioassays to Assess Ecotoxicological Impact on Soils Affected by Mining Activities in the Iberian Pyrite Belt"

_toxics, 2022, doi:10.3390/toxics10070353_

Round 1
Reviewer 1 Report
Dear authors,
The manuscript needs extensive revision for language and grammar.
Review the following: In Table 3 and for the assay of the inhibition growth of P. subcapitata, you mention that soil (samples) are not contaminated. You cannot know this. According with ANNEX IV “Criteria for identifying sites that require risk assessment” of the RD 9/2005: “2.c) Where toxicity is demonstrated based on any of the biotests referred to in Annex I II.2, carried out using soil or leachate in undiluted samples”. Therefore, you should mention that these soil (samples) are potentially contaminated soils and a risk assessment is needed to know if the soil is contaminated or not.
The following statement should be changed in Tables and text. “** R.D. 9/2005 Classification: EC50 < 1% C (Contaminated); EC50 > 1% N.C. (Not Contaminated)”
Do you think that the number of soil samples are enough to take your conclusions?
Lines 2-3: Please check the title: capital letters...
Line 19: I wonder if “was to use of” is correct in English.
Line 24: “shows” or “showed”
Line 29: please, review English: “a similar to”?
Line 31: “PHEs”à potentially harmful elements (PHEs)
Lines 53-58: The following paragraph is not clear: “In order to assess the toxicity of a group of elements in the different environmental compartments, total content of contaminants do not provide enough information, and even leachable or (bio)available fraction without the effect in the ecosystems, the toxicity in a real environmental condition”.
Line 80: “has provide”à”has provided”
Line 81: “metal content (0.7Cu, 0.8Pb and 1.4Zn)”àDo these numbers need a symbol unit?
Line 83: “the Lousal mine has”à review verbal tense
Line 85-86: “a large volume of mining waste has been produced”à review verbal tense
Line 90: Check the English of “An ore deposits”
Line 92: Check the English of “when the mining production has ceased in 1988”
Line 97: Check the English of “remediation consists in”
Line 104: “soils characterized by [7]”à soils characterized by Sánchez-Donoso et al. [7].
Line 120: Please, add subscripts to “SO3” and “Fe2O3”.
Line 141: Please review all the manuscript and change “ºC” by “ºC”
Line 142: “(Pall®), the water extract is then”à “(Pall®). The water extract was then”
Line 148: “(Microbiotests Inc, Belgium”à”(Microbiotests Inc, Belgium)”
Line 165: Please, check “Five dilutions: 50, 25. 12.5”
Line 174: Please, check “(μ)”
Line 189: Remove the abbreviation “WHC”.
Line 222: “proposed by [27]”à” proposed by Persoone et al. [27]”. In this sense, check line 337 “developed by [27],”
Line 276: “48h”à “48 h”. The same for “72h”, “7d” (line 292)…check also figures.
Line 294: “OECD207”à “OECD TG 207”?
Line 323: Write well the EC50
Check all the manuscript: EC50, EC50.
Line 339: “(Class V). ii)”à “(Class V); ii)”
Line 342-343: “D. magna”à italics
Line 365: Please, add the meaning of the X-axis
Line 429: Review “Roca-Péz, L. and Ruiz-Costa, N. and Roca-Perez, L”
Author Response
All the manuscript has been revised for language and grammar
Review the following: In Table 3 and for the assay of the inhibition growth of P. subcapitata, you mention that soil (samples) are not contaminated. You cannot know this. According with ANNEX IV “Criteria for identifying sites that require risk assessment” of the RD 9/2005: “2.c) Where toxicity is demonstrated based on any of the biotests referred to in Annex I II.2, carried out using soil or leachate in undiluted samples”. Therefore, you should mention that these soil (samples) are potentially contaminated soils and a risk assessment is needed to know if the soil is contaminated or not.
The Ministry of the Environment, in its effort to clarify the evaluation criteria of RD 9/2005, published in 2007 the Technical Guide (TG) for the Application of RD 9/2005 (prepared by the Geological and Mining Institute and the National Institute for Agricultural Research and Technology),
The Technical guide is available at: https://www.miteco.gob.es/es/calidad-y-evaluacion-ambiental/temas/suelos-contaminados/guia_tecnica_contaminantes_suelo_declaracion_suelos_tcm30-185726.pdf
This TG in its Annex III (page 54, last paragraph) establishes that "For the protection of ecosystems, the declaration of a soil as contaminated is based exclusively on ecotoxicity tests. Thus, unlike the criteria for the protection of human health, these criteria do not rely on NGRs, but on ecotoxicity data obtained from bioassays carried out on soil and leachate samples at different concentrations. This allows the classification of problematic sites without the need for a comprehensive characterization of the types and levels of present contaminants".
On the other hand, the TG (page 56, box 1) also establishes that the Annex IV this annex shall only be applicable for soils which fulfil one of the three conditions, i.e. i) contamination by Hydrocarbons, ii) contamination by organic compounds listed in Annex V (priority substances) or iii) when there is evidence that the levels of any pollutant are higher than the reference generic levels (NGR) established according to Annex VII on the basis of soil uses. As you can see, the mining is not included as such and therefore it is not possible to calculate the generic levels.
Finally, Annex III of the TG (page 54, box2 and page 55, last paragraph) states that: “A soil may be declared as contaminated if the L(E)C50 value for the most sensitive taxonomic group of soil is less than 10 mg contaminated soil/g soil (1% w/w) or if the L(E)C50 value of the leachate for the most sensitive taxonomic group of aquatic organisms is less than 10 ml leachate/l water (1% v/v”). In the general soil classification, it makes no difference whether the soil is uncontaminated for R. subcapitata or uncontaminated for Daphnia magna, in which case the worst case prevails.
The following statement should be changed in Tables and text. “** R.D. 9/2005 Classification: EC50 < 1% C (Contaminated); EC50 > 1% N.C. (Not Contaminated)”
For the above reasons, we believe that this statement should remain
All the following considerations have been considered and the text has been modified:
Lines 2-3: Please check the title: capital letters...
Line 19: I wonder if “was to use of” is correct in English.
Line 24: “shows” or “showed”
Line 29: please, review English: “a similar to”?
Line 31: “PHEs”à potentially harmful elements (PHEs)
Lines 53-58: The following paragraph is not clear: “In order to assess the toxicity of a group of elements in the different environmental compartments, total content of contaminants do not provide enough information, and even leachable or (bio)available fraction without the effect in the ecosystems, the toxicity in a real environmental condition”.
Line 80: “has provide”à”has provided”
Line 81: “metal content (0.7Cu, 0.8Pb and 1.4Zn)”àDo these numbers need a symbol unit?
Line 83: “the Lousal mine has”à review verbal tense
Line 85-86: “a large volume of mining waste has been produced”à review verbal tense
Line 90: Check the English of “An ore deposits”
Line 92: Check the English of “when the mining production has ceased in 1988”
Line 97: Check the English of “remediation consists in”
Line 104: “soils characterized by [7]”à soils characterized by Sánchez-Donoso et al. [7].
Line 120: Please, add subscripts to “SO3” and “Fe2O3”.
Line 141: Please review all the manuscript and change “ºC” by “ºC”
Line 142: “(Pall®), the water extract is then”à “(Pall®). The water extract was then”
Line 148: “(Microbiotests Inc, Belgium”à”(Microbiotests Inc, Belgium)”
Line 165: Please, check “Five dilutions: 50, 25. 12.5”
Line 174: Please, check “(μ)” the OECD TG 201 the abbreviation of “growth rate” term is μ
Line 189: Remove the abbreviation “WHC”.
Line 222: “proposed by [27]”à” proposed by Persoone et al. [27]”. In this sense, check line 337 “developed by [27],”
Line 276: “48h”à “48 h”. The same for “72h”, “7d” (line 292)…check also figures.
Line 294: “OECD207”à “OECD TG 207”?
Line 323: Write well the EC50
Check all the manuscript: EC50, EC50.
Line 339: “(Class V). ii)”à “(Class V); ii)”
Line 342-343: “D. magna”à italics
Line 365: Please, add the meaning of the X-axis
Line 429: Review “Roca-Péz, L. and Ruiz-Costa, N. and Roca-Perez, L”
Reviewer 2 Report
1) the manuscript should be read by a fluent English speaker to correct some inaccuracies in meaning eg
you state have produced an important contamination
bad english please rephrase
replace
being the test with Daphnia magna the most sensitive and 26 the Eisenia foetida the least sensitive.
with
The test with Daphnia magna was the most sensitive bioassay while and 26 the Eisenia foetida test was the least sensitive.
replace
showed a similar to that found in direct ecotoxicological bioassays
with
showed results similar to the ecotoxicological bioassays
you state
Mining environments suppose a risk for metals and metalloids dissemination from 36 the ore to the environmental compartments bad english please rephrase
you state In some areas, poorly devel-74 oped soils over the Alvalade basin detritic deposits are also present, characterized by a 75 sandy texture, very limited organic matter content, and a slightly acidic nature
a verb is missing somewhere etc...
2) it is not clear to me why a cell line assay was included to the ecotoxicity assays. you have top justify this
3) in the abstract you do not give information about results of mineralogical and geochemical research but only on the ecotoxicity tests. please include these also
4) in the introduction in the final paragraph please give information on: the problem, how you analyze it, why this is important for an international audience, how the results can be utilized
5) you state The most common soils are Leptosols
why in capital letter?
6) you state (0.7Cu, 0.8Pb and 1.4Zn)
what are these numbers?
7) you use abbreviations eg AMD but you havent explained what they mean
8) In figure 1 I dont understand what the sites are (I think you have not presented what "dump", "aerobic pond" mean). are these site used here or in the other paper mentioned? in any case you have to explain what these sites mean. also were the samples taken before the restoration programme or after? or both? it is not clear to me
9) in the materials and methods for all apparatuses give name, company name, city, country of origin. please see relevant papers in toxics for how to report on the methods. this is very important on the validity of the paper
10) an important issue in the experimental design is that in some experiments you use leachates whereas in the earthworm test you use the actual soil. you have to justify why there is this discrepancy. also you have to link the results to each other given that the substrate (soil vs leachate) is different
11) in the statistical analysis I dont understand why you used 2 way ANOVA what are the parameters (2) that you tested in the 2 way ANOVA?
12) In table 1 what % means? w/w? why there are also % and mg/kg? is table 1 analyzed at all in the main text? please elaborate. the results shown in 246-258 are also shown somewhere in some table? or any photos available?
13) I am not sure from the results-can you calculate a % of leachability based on the actual concentration of the soil compared to the concentration found in the leachate (given that you know how many grams of soil you subjected to leachate?). this would be very interesting. also you cannot claim that As was not leaching unless you compare to the initial As found in the soil
14) as stated before, I cannot see where you used 2 way anova. From the ecotoxicty data I cannot understand what you compared to what (probably from one way ANOVA as it seems)-each group to other in the same time point? or was time used as the second parameter in a two-way ANOVA? please explain better the statistics in the ecotoxicity assays
15) I am not sure how and where you used the results from mineralogy etc in your discussion-how were these data linked to the ecotoxicity results you detected? please elaborate more in the discussion and add relevant references. please see and quote
,
Science of The Total Environment,
Volume 655,
2019,
Pages 1457-1467
Author Response
All the suggestions made by this referee have been considered, We are very grateful for its recommendations.
The text has been modified and all the minor mistakes have been corrected.
It is not clear to me why a cell line assay was included to the ecotoxicity assays. you have top justify this.
As regards the cell line, the assay has been applied in order to approximate the human risk assessment. Particularly, the skin is the first barrier, and for this raison a skin assay is a good approximation to the real risk of PHEs. The other bioassays have been selected for ecosystems risk assessment.
In the abstract you do not give information about results of mineralogical and geochemical research but only on the ecotoxicity tests. please include these also
The abstract section has been modified and geochemical and mineralogical results have been included in this section.
In the materials and methods for all apparatuses give name, company name, city, country of origin. please see relevant papers in toxics for how to report on the methods. this is very important on the validity of the paper
The text has been modified and the name of apparatuses have been included.
An important issue in the experimental design is that in some experiments you use leachates whereas in the earthworm test you use the actual soil. you have to justify why there is this discrepancy. also you have to link the results to each other given that the substrate (soil vs leachate) is different
Each bioassay needs a different material. Is not a discrepancy.
Reviewer 3 Report
Dear Authors,
In my opinion the theme of the article is very actual and interesting for the readers of the journal.
The manuscript used geochemical, mineralogical and biological indicators for a complete assessment of ecological risks related to the mobility, ecotoxicity and bioavailability of potentially harmful elements in Lousal mining district. The authors should include in title the name of the country, where the mine was explored.
The geochemical and mineralogical characterization of the studied area shows that the mine waste dumps have produced an important contamination of the adjacent soils and nearby watercourses.
Particularly, the toxicity was evaluated by using four assays: algae, cytotoxicity assays with HaCaT cell line (dermal), earthworms and Daphnia magna.
The ecotoxicity bioassays was performed in selected soils and showed different sensitivities to the target contaminants. The test with the most sensitive .
Except for LOS07 soil sample, the rest of soils were classified as "High acute toxicity" and "Very high acute toxicity" for aquatic systems.
The results in HACaT cells showed a similar to that found in direct ecotoxicological bioassays.
In conclusion, the application of biotests together with the geochemical and mineralogical characterization showed to be a very useful tool to establish the degree of contamination and the environmental risk of PHEs
The paper is well structured, well written, the language is correct and clear, and the title shoul be clearify, and abstract clearly describe the content of the manuscript.
The authors must remove references in the Conclusion Section.
In my opinion minor revision is needed. Please see attached file.
Best regards

Author Response
The authors are grateful to this referee for its valuable suggestions. All of them have been included in the text
Round 2
Reviewer 1 Report
Dear authors,
I return to the comments I made in the previous revision.
“Review the following: In Table 3 and for the assay of the inhibition growth of P. subcapitata, you mention that soil (samples) are not contaminated. You cannot know this. According to ANNEX IV “Criteria for identifying sites that require risk assessment” of the RD 9/2005: “2.c) Where toxicity is demonstrated based on any of the biotests referred to in Annex I II.2, carried out using soil or leachate in undiluted samples”. Therefore, you should mention that these soil (samples) are potentially contaminated soils and a risk assessment is needed to know if the soil is contaminated or not”
In your answer, you mention: “On the other hand, the TG (page 56, box 1) also establishes that the Annex IV this annex shall only be applicable for soils which fulfil one of the three conditions, i.e. i) contamination by Hydrocarbons, ii) contamination by organic compounds listed in Annex V (priority substances) or iii) when there is evidence that the levels of any pollutant are higher than the reference generic levels (NGR) established according to Annex VII on the basis of soil uses. As you can see, the mining is not included as such and therefore it is not possible to calculate the generic levels”
At least, ii) is for human health, however, if I understand correctly you are considering ecological risk.
I wonder why you do not mention TG (page 57, box 2) or ANNEX IV 2.c). In this page, it is also mentioned (in the language of the TG) “Por tanto, en aquellos casos en que se considere prioritaria la protección de los ecosistemas, los suelos requerirán una valoración de riesgo, si se cumple alguna de las siguientes condiciones:
- Se observa toxicidad en los bioensayos realizados con las muestras de suelo o lixiviados sin diluir para los organismos del suelo y acuáticos mencionados en el anexo III"
” Therefore, in those cases in which the protection of ecosystems is considered a priority, the soils will require a risk assessment, if any of the following conditions are met:
- Toxicity is observed in the bioassays carried out with soil samples or undiluted leachate for soil and aquatic organisms mentioned in Annex III”
Author Response
Review the following: In Table 3 and for the assay of the inhibition growth of P. subcapitata, you mention that soil (samples) are not contaminated. You cannot know this. According with ANNEX IV “Criteria for identifying sites that require risk assessment” of the RD 9/2005: “2.c) Where toxicity is demonstrated based on any of the biotests referred to in Annex I II.2, carried out using soil or leachate in undiluted samples”. Therefore, you should mention that these soil (samples) are potentially contaminated soils and a risk assessment is needed to know if the soil is contaminated or not.
The following statement should be changed in Tables and text. “** R.D. 9/2005 Classification: EC50 < 1% C (Contaminated); EC50 > 1% N.C. (Not Contaminated)”
I return to the comments I made in the previous revision.
“Review the following: In Table 3 and for the assay of the inhibition growth of P. subcapitata, you mention that soil (samples) are not contaminated. You cannot know this. According to ANNEX IV “Criteria for identifying sites that require risk assessment” of the RD 9/2005: “2.c) Where toxicity is demonstrated based on any of the biotests referred to in Annex I II.2, carried out using soil or leachate in undiluted samples”. Therefore, you should mention that these soil (samples) are potentially contaminated soils and a risk assessment is needed to know if the soil is contaminated or not”
In your answer, you mention: “On the other hand, the TG (page 56, box 1) also establishes that the Annex IV this annex shall only be applicable for soils which fulfil one of the three conditions, i.e. i) contamination by Hydrocarbons, ii) contamination by organic compounds listed in Annex V (priority substances) or iii) when there is evidence that the levels of any pollutant are higher than the reference generic levels (NGR) established according to Annex VII on the basis of soil uses. As you can see, the mining is not included as such and therefore it is not possible to calculate the generic levels”
At least, ii) is for human health, however, if I understand correctly you are considering ecological risk.
I wonder why you do not mention TG (page 57, box 2) or ANNEX IV 2.c). In this page, it is also mentioned (in the language of the TG) “Por tanto, en aquellos casos en que se considere prioritaria la protección de los ecosistemas, los suelos requerirán una valoración de riesgo, si se cumple alguna de las siguientes condiciones:
- Se observa toxicidad en los bioensayos realizados con las muestras de suelo o lixiviados sin diluir para los organismos del suelo y acuáticos mencionados en el anexo III"
” Therefore, in those cases in which the protection of ecosystems is considered a priority, the soils will require a risk assessment, if any of the following conditions are met:
- Toxicity is observed in the bioassays carried out with soil samples or undiluted leachate for soil and aquatic organisms mentioned in Annex III”
Thank you for your valuable comments, we believe they are helpful for the improvement of the manuscript. The omission of the point TG (page 57, box 2) or ANNEX IV 2.c was not intentional. We believe that the discrepancy is due to the interpretation of the concept "priority protection of ecosystems". According to RD 9/2005, in article 4.2: The competent body of the autonomous community will delimit those soils in which the protection of the ecosystem of which they are part is considered a priority. In each of these cases, the competent authority will determine which taxonomic group/s of organisms should be protected". In this case, there is no evidence that the Lousal area has been declared as “priority for ecosystem protection” (the study location is a mining area not a natural protected area, natural reserve or special protection area). Therefore, the provisions of Annex IV regarding "Criteria for identifying sites that require risk assessment" would not apply. Consequently, the characterization of the soil as “contaminated” or “Not contaminated” should be made on the basis of EC50s obtained in each bioassay.
In order to clarify this aspect in the manuscript, we have added a sentence stating: “if ecosystem protection is declared by the local Authority, a complementary environmental risk assessment study should be carried out in accordance with Annex VI of RD9/2005.
Reviewer 2 Report
I am sorry but unless I missed some parts of your revision, you have not answered my remarks. you have answered up to no. 10 whereas I had more comments. also not all remarks up to no 10 were answered and some were answered very superficially eg that the cell assay was done because the skin is in contact with the pollutants? skin of whom? I believe you dealt with ecotoxicology not human toxicology. I am sure that there is an explanation of your choice of tests but this was not communicated to me. I understand that maybe you needed more time to do the corrections so you should have asked so. I cannot consent to approval of the manuscript
Author Response
The authors are very grateful to this referee. We have responded all questions. Sorry, in the first revision we had a problem with the document.
1) The manuscript should be read by a fluent English speaker to correct some inaccuracies.
Thanks for the suggestion, the manuscript has been revised for the language.
Replace “being the test with Daphnia magna the most sensitive and 26 the Eisenia foetida the least sensitive” with “The test with Daphnia magna was the most sensitive bioassay while and 26 the Eisenia foetida test was the least sensitive”.
We agree, please, note that the text has been modified.
Replace “showed a similar to that found in direct ecotoxicological bioassays” with “showed results similar to the ecotoxicological bioassays”
We agree, please, note that the text has been modified.
Mining environments suppose a risk for metals and metalloids dissemination from 36 the ore to the environmental compartments bad english please rephrase
The introduction section has been slightly modified.
you state: In some areas, poorly developed soils over the Alvalade basin detritic deposits are also present, characterized by a sandy texture, very limited organic matter content, and a slightly acidic nature
We are grateful to this suggestion; the verb in this sentence is “are present”.
2) it is not clear to me why a cell line assay was included to the ecotoxicity assays. you have top justify this
Ecotoxicological tests are aimed to evaluate the (eco)toxic effect on ecosystems. Toxicological tests do so for animals (including humans). In the latter, the use of cell lines to assess cytotoxicity is the first step in the process of studying substances on animals.
Based on the results of the ecotoxicity bioassays (algae, daphnia and earthworms) it was determined an environmental risk. In this sense, it was decided to extend the risk assessment to a higher tier (the potentially effect on human). We posed to study the potential effect of the substances present in the soils against a cellular model. In this sense, considering that the cutaneous route is one of the most important routes of entry of chemical substances into the human organism a dermal cell line (HaCaT) was chosen.
The use of ecotoxicological assays in the evaluation of mining contamination is quite novel in its approach. We believe that cytotoxicity assays with human cells are a very good tool to evaluate the effect on vertebrate animals (including humans) at a first level of complexity. The study with HaCaT cell line is aimed at assessing the potential risk to workers or nearby population exposed to particulate matter from the study area either by direct contact (e.g. dust) or by indirect route (e.g. natural water contamination due to leaching process). Because the cells must be in a liquid growth medium, it is not possible to use soil, so leachate (soluble fraction) was used.
3) in the abstract you do not give information about results of mineralogical and geochemical research but only on the ecotoxicity tests. please include these also.
The abstract section has been modified and a sentence about mineralogical and geochemical results has been included.
4) in the introduction in the final paragraph please give information on: the problem, how you analyze it, why this is important for an international audience, how the results can be utilized
The aim of the study has been better explained and they modification is now included in the manuscript.
5) you state The most common soils are Leptosols, why in capital letter?
There was a mistake, the text has been modified.
6) you state (0.7Cu, 0.8Pb and 1.4Zn), what are these numbers?
Please note that this numbers have been deleted.
7) you use abbreviations e.g. AMD but you haven’t explained what they mean
AMD stands for Acid Mine Drainage. The first time that the abbreviation appears in the text has been specified
8) In figure 1 I dont understand what the sites are (I think you have not presented what "dump", "aerobic pond" mean). are this site used here or in the other paper mentioned? in any case you have to explain what these sites mean. also were the samples taken before the restoration programme or after? or both? it is not clear to me
Done. A brief explanation about dump and aerobic wetland was added to the paragraph, but most of the information can be completed in Sanchez-Donoso et al. Samples were taken before the restoration works. The new paragraph is as follows: “Samples used in this study correspond to a selection of soils characterized by Sánchez-Donoso et al. [7]. Selection criteria were based on contrasting PHEs concentrations. As it can be seen in figure 1, the area has a dump with pyrite wastes upstream and a AMD passive treatment (aerobic wetland) for the acid waters. Samples were collected from the mine soils in the restoration area before the works began. As Figure 1 displays, most of the samples (LOS06 - LOS10) were taken in the west transect, in the flow direction downstream the pyrite wastes (dump in figure 1), while the last sample (LOS13) corresponds to the lower one of the East transect, close to the aerobic wetland treatment.”
9) in the materials and methods for all apparatuses give name, company name, city, country of origin. Please see relevant papers in toxics for how to report on the methods. This is very important on the validity of the paper.
The name of the apparatuses is now included in the manuscript and this section has been revised and modified according to similar previous works.
10) an important issue in the experimental design is that in some experiments you use leachates whereas in the earthworm test you use the actual soil. you have to justify why there is this discrepancy. Also you have to link the results to each other given that the substrate (soil vs leachate) is different
According to the OECD Technical Guideline followed in the study, each bioassay was performed on the matrix for which its application is indicated. Eisenia foetida is a terrestrial invertebrate living in soil, therefore the whole soil samples have been used. Daphnia magna is a freshwater microcrustacean (commonly known as waterflea) and Raphidocelis subcapitata is a freshwater green algae. Both require an aquatic environment, in this sense the OECD guidelines state that the leachate generated from the soil sample should be used. In all bioassays, ecotoxicity is expressed as EC50, which is the concentration (of soil or leachate) that produces an effect in 50% of the population tested at a given exposure time (i.e mortality in Daphnia magna and Eisenia foetida tests and growth inhibition in the case of R. subcapitata). This EC50 value, calculated using a standardized accepted method (Probit regression), equals the toxicity values, independently of the matrix applied.
11) in the statistical analysis I don’t understand why you used 2 way ANOVA what are the parameters (2) that you tested in the 2 way ANOVA?
It was a typo; the statistical analysis was based in an One-way ANOVA as stated in the reference OECD TG. Only one parameter is compared.
12) In table 1 what % means? w/w? why there are also % and mg/kg? is table 1 analyzed at all in the main text? please elaborate. The results shown in 246-258 are also shown somewhere in some table? or any photos available?
Major results are expressed in %wt and minor elements in mg/kg. In addition, a new table, including mineralogical results, has been included.
13) I am not sure from the results-can you calculate a % of leachability based on the actual concentration of the soil compared to the concentration found in the leachate (given that you know how many grams of soil you subjected to leachate?). this would be very interesting. also you cannot claim that As was not leaching unless you compare to the initial As found in the soil
Taking into account that some bioassays are carried out in leachates, the obtained results are compared with the most mobile fraction, the soluble one. This fraction has been calculated with a 1:5 soil:water extract.
14) as stated before, I cannot see where you used 2 way anova. From the ecotoxicity data I cannot understand what you compared to what (probably from one way ANOVA as it seems)-each group to other in the same time point? or was time used as the second parameter in a two-way ANOVA? please explain better the statistics in the ecotoxicity assays
Thanks for your valuable comment. You are true, as stated above there was a typo. The statistic procedure was a One-Way ANOVA. The comparation was made for the “% leachate” vs. Control inside each sampling point. This has been better explained in the figure caption
15) I am not sure how and where you used the results from mineralogy etc in your discussion-how were these data linked to the ecotoxicity results you detected? please elaborate more in the discussion and add relevant references. please see and quote Environmental Science and Pollution Research this link is disabled, 2021, 28(4), pp. 3797–3809 and Science of The Total Environment, Volume 655, 2019, Pages 1457-1467
The ecotoxicity tests results have been analyzed and discussed in detail, since aspects related to PTEs content and mineralogical composition have been discussed in a previous work carried out by the authors (Sánchez-Donoso et al., 2021).
The suggested citations are very interesting, although the pollutants analyzed in these references behave differently from PTEs. There are many published works on the use of bioassays and risk assessment procedure, which is why we have used bibliography related to mining areas and PTEs contamination.
Round 3
Reviewer 1 Report
Dear authors,
Now, it is clear the topic of the "priority protection of ecosystems".
A minor change has to be made in the manuscript. If you agree with me, I would change “The RD establishes as “contaminated soils” those with an EC50 < 1% (in accordance with RD Annex IV, if the protection of the ecosystem is considered a priority by the local Authorities – i.e. declared a protected natural area – an environmental risk assessment should be carried out)” by “The RD establishes as “contaminated soils” those with an EC50 < 1%. In accordance with RD Annex IV, if the protection of the ecosystem is considered a priority by the local Authorities – i.e. declared a protected natural area – and the EC50 is higher than 1%, an environmental risk assessment should be carried out”
Author Response
Thank you. The text has been modified.
Reviewer 2 Report
accepted
Author Response
Thank you